# Demonstration of Ultra-High-Q Silicon Microring Resonators for Nonlinear Integrated Photonics

**DOI:** 10.3390/mi13071155

**Published:** 2022-07-21

**Authors:** Desheng Zeng, Qiang Liu, Chenyang Mei, Hongwei Li, Qingzhong Huang, Xinliang Zhang

**Affiliations:** Wuhan National Laboratory for Optoelectronics, Huazhong University of Science and Technology, Wuhan 430074, China; zengdsh@hust.edu.cn (D.Z.); liuq@mail.hust.edu.cn (Q.L.); m201972867@hust.edu.cn (C.M.); d202181023@hust.edu.cn (H.L.); xlzhang@mail.hust.edu.cn (X.Z.)

**Keywords:** Q-factor, SOI microring resonator, four-wave mixing, conversion efficiency

## Abstract

A reflowing photoresist and oxidation smoothing process is used to fabricate ultra-high-Q silicon microring resonators based on multimode rib waveguides. Over a wide range of wavelengths near 1550 nm, the average Q-factor of a ring with 1.2-μm-wide waveguides reaches up to 1.17 × 10^6^, with a waveguide loss of approximately 0.28 dB/cm. For a resonator with 1.5-μm-wide waveguides, the average Q-factor reaches 1.20 × 10^6^, and the waveguide loss is 0.27 dB/cm. Moreover, we theoretically and experimentally show that a reduction in the waveguide loss significantly improves the conversion efficiency of four-wave mixing. A high four-wave mixing conversion efficiency of −17.0 dB is achieved at a pump power of 6.50 dBm.

## 1. Introduction

Silicon photonics has emerged as a promising photonic integration platform for compatible complementary metal-oxide-semiconductor (CMOS) fabrication technology [1,2,3,4]. Moreover, compact devices are usually desired and can be developed on silicon-on-insulator (SOI) platforms. Microring resonators (MRRs) are essential in a variety of applications, including optical filters [5,6,7], frequency combs [8,9,10], optical buffers [11,12], optical switches [13], and biosensors [14]. The Q-factor (Q) is a critical parameter of an MRR that represents the number of field oscillations before the circulating energy is depleted to 1/*e* of the initial energy. However, the Q of a silicon MRR is limited as a result of scattering loss caused by the roughness of the waveguide sidewall due to fabrication imperfections as well as bending loss [15].

To increase the value of Q, many researchers have used wider multimode waveguides in MRRs or racetrack resonators to reduce mode overlap with vertical sidewalls [16,17,18]. Guiller-Torres et al. obtained a Q of 1.7 × 10^6^ by using low-loss 3-μm-wide multimode straight rib waveguides and single mode strip waveguides in the bend regions [19]. In addition, a silicon racetrack resonator, composed of multimode long straight waveguides and single mode arc waveguides, has achieved a high Q value of 1.14 × 10^6^ and a wider free spectral range (FSR) of 0.325 nm [18]. While a silicon racetrack resonator based on an Euler multimode racetrack waveguide has demonstrated a Q value of approximately 1.3 × 10^6^ and an FSR of 0.9 nm. The use of Euler multimode waveguides can eliminate the need for tapered waveguides between single and multimode waveguides while suppressing the bending loss of circular waveguides [20]. The waveguide sidewalls can be smoothed with effective processing techniques, such as hydrogen annealing, laser burnishing, chemical mechanical polishing, reflowing photoresist (RP), and oxidation smoothing [21,22], to further minimize scattering loss. Junichi Takahashi et al. [23] obtained smooth waveguides with root mean square (RMS) roughnesses of 5–10, 3–5, 2–3, and 1–2 nm by oxidizing silicon waveguides at 1000 °C for 0, 18, 75, and 165 min, respectively. An AlGaAs-on-insulator nanowaveguide has used to achieve an ultra-high-Q MRR with a Q value of 3.52 × 10^6^ using RP and heterogeneous wafer bonding methods [24]. Shih-Che Hung et al. [25] utilized a KrF excimer laser to illuminate the surface of silicon waveguides, making the silicon material molten. As a result, the waveguides became smoother under surface tension. They succeeded in reducing the RMS of the waveguide roughness from 14 nm to 0.24 nm. However, laser irradiation is not easy to perform and requires expensive equipment. RP and oxidation smoothing techniques are simple and effective methods.

In this paper, we present and demonstrate ultra-high-Q silicon MRRs, which are composed of rib multimode straight waveguides and rib multimode microrings, to dramatically minimize the overlap between fundamental modes and the vertical sidewalls. To reduce scattering loss, we also use RP and oxidation smoothing techniques. Ultrahigh-Q MRRs with 1.2 μm- and 1.5 μm-wide rib silicon waveguides are fabricated. In addition, we analyze the coherent interference effect between fundamental and higher-order modes in the ring. The average loaded Qs for resonance wavelengths between 1500 nm and 1580 nm reach up to 1.17 × 10^6^ and 1.20 × 10^6^, respectively. Moreover, to demonstrate the high nonlinear efficiency of the proposed system, we investigate the degenerate four-wave mixing (FWM) [26] of ultra-high-Q MRRs. The experimentally measured maximum conversion efficiency (CE) of the degenerate FWM reaches up to −17.0 dB at a pump power of 6.50 dBm for an MRR with 1.5-μm-wide rib waveguides. The present silicon photonic resonators are expected to be widely used in many applications, including microwave photonic filters, optical sensors and integrated material platforms specifically for nonlinear photonics.

## 2. Structure and Fabrication

Figure 1 depicts the schematic configuration and characterization of the proposed ultra-high-Q silicon all-pass microring resonator (APMRR). The MRR and the bus waveguide have the same width. As shown in Figure 1a, the width and radius of the waveguides are W_rib_ and R, respectively. Moreover, the MRR supports two modes (Mode 1 and Mode 2). To reduce the scattering loss at the waveguide sidewalls, an oxide layer is grown on the shallow etched rib waveguides. As shown in Figure 1b, the rib height and the thickness of the thermal oxide layer are 70 nm and 50 nm, respectively. When the light from the bus waveguide is coupled laterally into the ring, two modes are excited, namely, the TE_0_ mode and TE_1_ mode, as shown in Figure 1a. The mode distribution of the TE_0_ mode is more strongly confined in the rib waveguide than that of the TE_1_ mode, and the overlap of the TE_0_ mode with the sidewall is smaller than that of the TE_1_ mode. According to Ref. [27], the fundamental mode of the bus waveguide can be evanescently and directly coupled to the fundamental mode or higher-order modes of the ring, whereas the fundamental mode and higher-order modes are indirectly coupled in multimode rings. The transfer function at the through port [28] of the APMRR can be described as:(1)$$T=\frac{t_{s}-\sum_{i=0}^{1}\alpha_{i}\left(k_{i}^{2}+t_{s}t_{i}\right)e^{j\varphi_{i}}+\alpha_{0}\alpha_{1}\left(2k_{0}k_{1}k_{c}+k_{0}^{2}t_{1}+k_{1}^{2}t_{0}-k_{c}^{2}t_{s}+t_{s}t_{0}t_{1}\right)e^{j(\varphi_{0}+\varphi_{1})}}{1-\sum_{i=0}^{1}\alpha_{i}t_{i}e^{j\varphi_{i}}+\alpha_{0}\alpha_{1}\left(-k_{c}^{2}+t_{0}t_{1}\right)e^{j(\varphi_{0}+\varphi_{1})}}$$
where −*jk*_0,1_ is the field coupling coefficient between the mode of the bus waveguide and TE_0,1_ mode of the ring; −*k_c_* is the field coupling coefficient between TE_0_ and TE_1_ modes in the ring; *t**_s_*_,0,1_ is the field transmission coefficients. Here, *φ*_0,1_ and *α*_0,1_ are the round-trip phase shift and round-trip field attenuation in the ring for TE_0_ and TE_1_ mode, respectively.

The devices were fabricated on a commercial SOI wafer with a 220 nm-thick top silicon layer and a 2-μm-thick buried oxide layer. A standard ARP6200.09 photoresist layer was spun and patterned using electron beam lithography. After the photoresist layer was carefully developed, the RP process was carried out on a hotplate at 130 °C for 1 min. Then, the patterns were transferred to the silicon layer using an inductively coupled plasma etcher. The remaining photoresist was then removed by dipping the wafer into a 1-methyl-2-pyrrolidinone stripper. Finally, a 50-nm-thick oxide layer was grown on the device via thermal oxidation. The specific process is that placing the silicon chip in an oxygen atmosphere at 950 °C and oxidizing it for 114 min. In the fabrication process of MRRs based on multimode waveguides, the RP and oxidation smoothing methods are both critical steps. The RP method forms a smoother etch mask pattern and reduces the roughness caused by plasma dry etching. The reflowing time should be carefully controlled. Figure 2a shows a metallographic micrograph of a ring resonator with a radius of 800 μm. To examine the reflow effect, we directly compare the patterns formed without and with the RP process before etching and show scanning electron microscope (SEM) images in Figure 2b,c. The edges of the photoresist patterns formed with RP shown in Figure 2c are smoother than the edges of the photoresist patterns formed without RP shown in Figure 2b, demonstrating that the roughness is not transferred to the waveguides during dry etching. Figure 2d,e show high-resolution top-down SEM graphs of the etched waveguides. The etched waveguide with RP shown in Figure 2e is smoother than the etched waveguide without RP shown in Figure 2d. Additionally, we use the commercial software ProSEM to calculate the RMS roughness (σ) based on the line-edge roughness profile extracted from the SEM images. The σ values of the etched waveguides with and without RP are 1.21 nm and 2.54 nm, respectively.

After the silicon waveguides are etched, oxidation smoothing is performed. This process can eliminate Si dangling bonds and flatten convex points on the waveguide surface because convex points react to the oxidant faster than the concave areas, reducing the roughness and scattering loss of the waveguide top surfaces and sidewalls. Notably, thermal oxidation transforms silicon into silica and reduces the height of the rib waveguide; thus, the oxidation time should be precisely controlled based on the actual situation.

## 3. Measurement and Analysis of Q

To characterize the fabricated ultra-high-Q APMRRs, the step size of the tunable laser source is set to 0.1 pm. The light from the tunable laser is adjusted with a polarization controller, and a vertical coupling system with grating couplers is used for efficient fiber-chip coupling. An optical power meter is used to record the transmission of the ultra-high-Q MRRs. The measured spectral response of an APMRR with a 1.2-μm-wide rib waveguide is shown in Figure 3a, demonstrating that the MRR has two resonance modes. The free spectral ranges (FSRs) of the two modes are 0.127 nm and 0.124 nm. The group indices (n_g_) of the TE_0_ and TE_1_ modes are 3.75 and 3.86 at 1550 nm, respectively. These values were substituted into the equation *FSR* = *λ*^2^/*n_g_L*, where *λ* and *L* are the wavelength in free space and the length of the resonator, respectively, and the calculated results are consistent with the measurements. Figure 3a shows that a fluctuating envelope for the TE_0_ mode resonance peaks varies with the spacing of the gap between the TE_0_ and TE_1_ mode resonance peaks. The reason for this phenomenon is the periodic variation in the coherent interference between the TE_0_ and TE_1_ modes. When the spacings between the TE_0_ and TE_1_ mode resonance peaks gradually increase, the impact of coherent interference on the Qs of the TE_0_ mode decreases. When the spacings between the TE_0_ and TE_1_ mode resonance peaks decrease, the intensity of the coherent interference increases, the TE_0_ mode partially converts into the TE_1_ mode, and the corresponding loss increases; thus, the coherent interference effects lead to a decrease in the value of Q. 

A Lorentzian fit to the output spectrum was performed, and parameters such as the field coupling coefficients, transmission coefficients, round-trip phase shifts and round-trip field attenuations of the fundamental mode and first-order mode were obtained by solving the Q and extinction ratio (ER) equations. For weak coupling, the field coupling coefficients for the fundamental mode and first-order mode with the bus waveguide were found to be 0.021 and 0.058, respectively, and the corresponding round-trip field attenuation coefficients were found to be 0.984 and 0.501. For strong coupling, a similar method was used to calculate the associated parameters. Then, by substituting the calculated results into Equation (1), we could obtain the spectral curves for weak coupling and strong coupling. The values of these parameters obtained from the experimental results were also substituted into Equation (1), and the spectral responses at typical resonance peaks with strong and weak coherent interference (i.e., weak coupling and strong coupling, respectively) are shown in Figure 3b,c, from which it can be seen that the spectra are consistent with the numerically calculated results. Notably, MRRs with two adjacent modes can enable efficient nonlinear conversion in some cases [29].

The histograms in Figure 4 show the Q values and waveguide losses of two ultra-high-Q MRRs with different rib waveguide widths at wavelengths between 1500 nm and 1580 nm. Considering the influence of coherent interference on Q, we count only Qs at weak couplings with spacings of greater than 0.045 nm between the TE_0_ and TE_1_ modes. Figure 4a,b show that the average Qs of these MRRs with 1.2- and 1.5-μm-wide waveguides are 1.17 × 10^6^ and 1.20 × 10^6^, respectively. For these two MRRs, the waveguide losses were calculated by solving the numerical Q-factor and ER equations. For the resonator with a waveguide width of 1.2 μm, the average waveguide loss is 0.28 dB/cm. For the resonator with a waveguide width of 1.5 μm, the average waveguide loss is 0.27 dB/cm. Thus, increasing the waveguide width is an effective method of improving the Q value of an MRR and reducing the scattering loss caused by sidewall roughness. Hence, RP and oxidation smoothing are effective methods for increasing the smoothness of a waveguide. In addition, the Q values at approximately 1550 nm are on the order of 10^5^ before oxidation smoothing and in-crease to the order of 10^6^ after oxidation smoothing.

## 4. Measurement of FWM

FWM is a nonlinear parametric process that has broad applications ranging from all-optical switching [30] and multiwavelength broadcasting [31] to correlated/entangled photon pair generation in quantum optical systems [32].

Continuous-wave degenerate FWM with an ultra-high-Q APMRR was investigated to demonstrate the efficiency of this nonlinear process. Figure 5a shows the experimental setup. The output wavelengths of the pump light and signal light should be carefully aligned to the resonance wavelengths of the ultra-high-Q resonator [33]. The pump light amplified by the erbium-doped fiber amplifier (EDFA) and the signal light are both modulated to the TE modes with a polarization controller (PC). The two TE mode lights are then combined with a 3 dB coupler and coupled into the ultra-high-Q MRR through a vertical grating coupler. The pump and signal wavelengths are 1543.24 nm and 1543.49 nm, respectively. Under low power input, the extinction ratios surrounding the pump and signal wavelengths exceed 10 dB, and the maximum Q value reaches up to 1.51 × 10^6^, as shown in Figure 5b. The degenerate FWM response of the resonator was measured by an optical spectrum analyzer (OSA). Figure 5c shows the output spectrum of the degenerate FWM of a resonator with 1.5-µm-wide rib multimode waveguides. When the power of the pump light is 6.5 dBm, an idler is observed, and the CE of the resonator reaches up to −17.0 dB. In addition, the conversion efficiency of degenerate FWM can be improved by carefully optimizing the dispersion and carrier extraction of the silicon MRR.

Because of the smaller overlap between the mode and the vertical sidewalls and the larger effective mode area, the multimode waveguide has lower linear and nonlinear losses than the single mode waveguide. Moreover, the larger effective mode areas in multimode waveguides reduce the waveguide nonlinearity parameter of FWM. Therefore, we theoretically investigated the impact of MRRs with different waveguide widths on the CE. The CE of FWM, *η*, in a MRR can be expressed as [34]:(2)$$\eta ={\left|\gamma PL_{eff}\right|}^{2}FE^{8}$$
(3)$$\gamma =\frac{n_{2}\omega }{cA_{eff}}$$
(4)$$L_{eff}=L\exp(-\alpha L/2)\left|\frac{1-\exp(-\alpha L+j\Delta kL)}{\alpha L-j\Delta kL}\right|$$
(5)$$FE=\left|\frac{\kappa }{1-\tau \exp(-\alpha L/2+jkL)}\right|$$
(6)$$\alpha =\alpha_{l}+\frac{\beta_{TPA}}{A_{eff}}P+\frac{\sigma_{F}\beta_{TPA}\tau_{eff}}{2hcA_{eff}{}^{2}}\lambda_{P}P^{2}$$
where *γ*, *P*, *L_eff_*, and *FE* are the nonlinearity parameter, pump power, effective length, and the field enhancement factor respectively, *n*_2_, *ω*, *A_eff_* are the nonlinear index coefficient, idler angular frequency, and effective area respectively, *L*, *α*, *κ* and *τ* are length of ring, the propagation attenuation coefficient, the ring coupling coefficient and transmission coefficient, respectively; Δ*k* represents 2*k_p_-k_s_-k_i_*, where *k_p_*_,*s*,*i*_ are the wavenumbers of the pump, signal, and idler fields, respectively; and *α_l_*, *β_TPA_*, *σ_FCA_*, *τ_eff_* and *λ_P_* are the linear attenuation coefficient, two-photon absorption coefficient, free carrier absorption area, carrier lifetime and pump wavelength, respectively. To investigate the impacts of the effective area and propagation attenuation on the CE, we selected rib waveguides with widths ranging from 0.5 µm to 1.8 µm, and fixed the linear loss to calculate the FWM CE. We applied the finite-difference method to calculate the effective areas and group velocity dispersions for various rib waveguides, with *n*_2_ = 6.5 × 10^−14^ cm^2^/W, *β_TPA_* = 0.7 cm/GW, *τ* = 0.5 ns, *σ_FCA_* = 1.45 × 10^−17^ cm^2^ [35,36,37], and a pump power of 6.5 dBm. When the changes in the effective area are taken into account, the solid lines in Figure 6 demonstrate that at the same coupling power, the CE decreases sharply as the linear loss increases. Although a larger effective area will result in a lower CE, the effect is weaker than the reduction in the waveguide loss. Thus, MRRs with multimode rib waveguides can achieve greater CEs by virtue of their decreased propagation loss, according to the calculation results shown as dots in Figure 6. Note that the linear loss coefficients of the waveguides are extracted from the experimental results.

We also measured the FWM CEs of MRRs with 1.2- and 1.5-µm-wide waveguides at different wavelengths. The results are presented as stars and are found to be consistent with the calculated results, which are shown as dots. Table 1 compares the CEs of degenerate FWM for various types of SOI MRRs and other material platforms, and the results indicate that multimode rings with wider waveguides can achieve higher FWM CEs than single mode rings can.

## 5. Conclusions

In summary, we have designed and demonstrated ultra-high-Q SOI MRRs based on multimode rib waveguides and have employed the RP and oxidation smoothing techniques to minimize the waveguide roughness and scattering loss. The average Qs of MRRs with 1.2- and 1.5-µm-wide waveguides at wavelengths between 1500 nm and 1580 nm are 1.17 × 10^6^ and 1.20 × 10^6^, corresponding to average waveguide losses of 0.28 dB/cm and 0.27 dB/cm, respectively. Furthermore, such an ultra-high-Q APMRR has an efficient FWM process, as demonstrated by degenerate FWM, which has a high CE of −17.0 dBm at a low pump power of 6.50 dBm. Furthermore, we have simulated the coherent interference between the TE_0_ and TE_1_ modes in the multimode ring, which results in periodic fluctuations in the TE_0_ mode resonance peaks. In the future, we will explore the effects of higher-order resonance modes in wavelength conversion in rings because of their ability to mitigate cavity thermo-optical effects. Our findings suggest that silicon-based multimode MRRs might be effective in improving the Q and CE of FWM.

## Figures and Tables

**Figure 1 micromachines-13-01155-f001:**
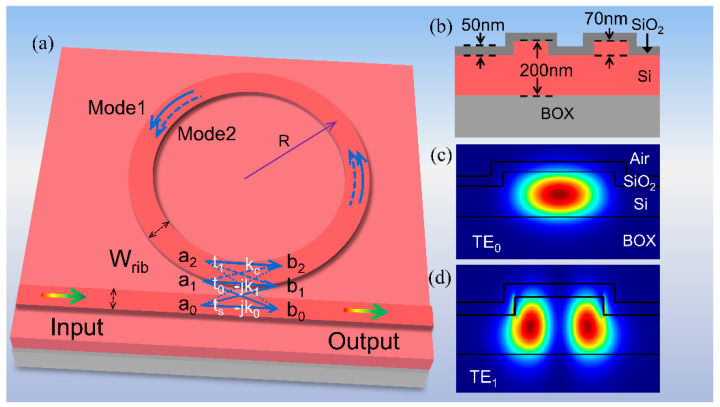
(**a**) 3D structure of a multimode MRR. Here, W_rib_ denotes the width of both the bus waveguide and the MRR. (**b**) The cross section of the coupling area. (**c**) The fundamental mode (TE_0_) distribution and (**d**) the first-order mode (TE_1_) distribution in the multimode rib waveguide.

**Figure 2 micromachines-13-01155-f002:**
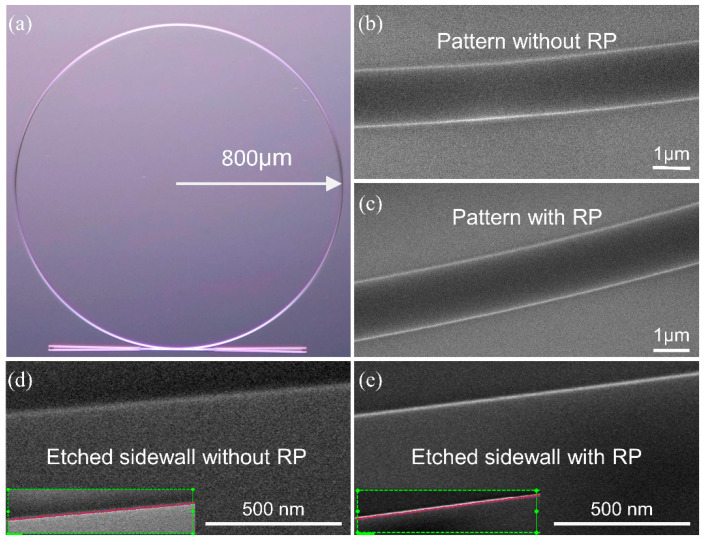
(**a**) The micrograph of ring resonator. Here, the radius of MRR is 800 μm. Top SEM view of photoresist patterns (**b**) without RP and (**c**) with RP. Top SEM view of etched waveguide sidewalls (**d**) without RP and (**e**) with RP. The inserts in (**d**,**e**) show the edge curve fitting results obtained in ProSEM.

**Figure 3 micromachines-13-01155-f003:**
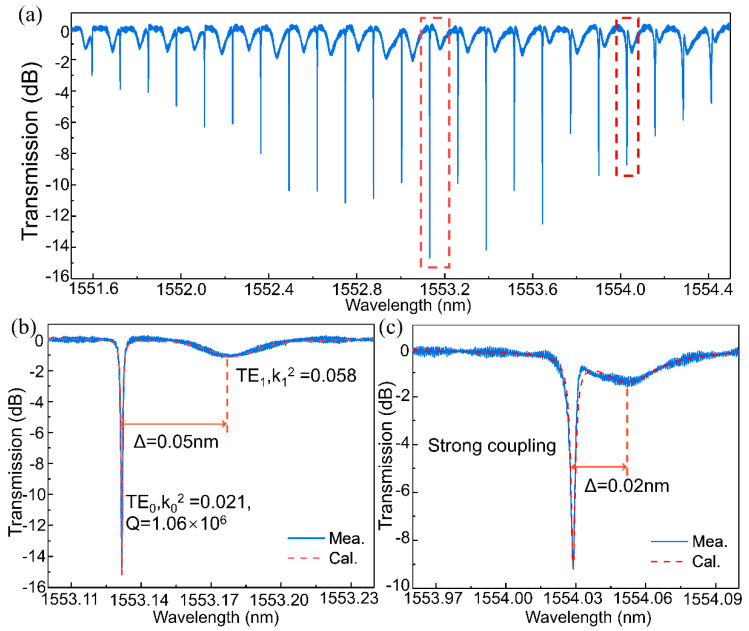
(**a**) The measured spectra at the through port of the ultra-high-Q MRR. Two whispering gallery modes (TE_0_ and TE_1_) are excited in the ring. The theoretical results are consistent with the experimental results for the TE_0_ and TE_1_ modes at the resonance peaks for (**b**) weak coupling and (**c**) strong coupling.

**Figure 4 micromachines-13-01155-f004:**
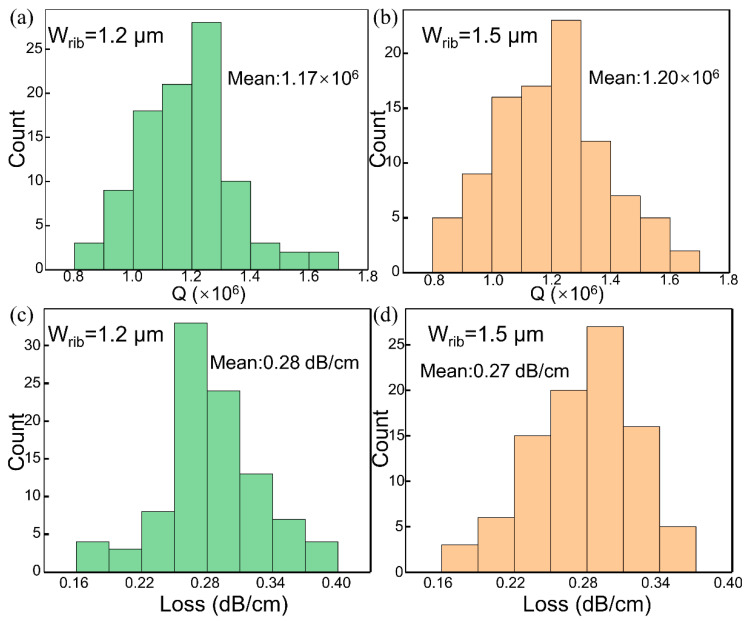
The statistics of the Q values and waveguide losses for ultra-high-Q MRRs between 1500 and 1550 nm. (**a**) For an MRR with a 1.2-μm-wide waveguide, the mean Q value is 1.17 × 10^6^. (**b**) For an MRR with a 1.5-μm-wide waveguide, the mean Q value is 1.20 × 10^6^. For MRRs with 1.2- and 1.5-μm-wide waveguides, the mean losses are (**c**) 0.28 dB/cm and (**d**) 0.27 dB/cm, respectively.

**Figure 5 micromachines-13-01155-f005:**
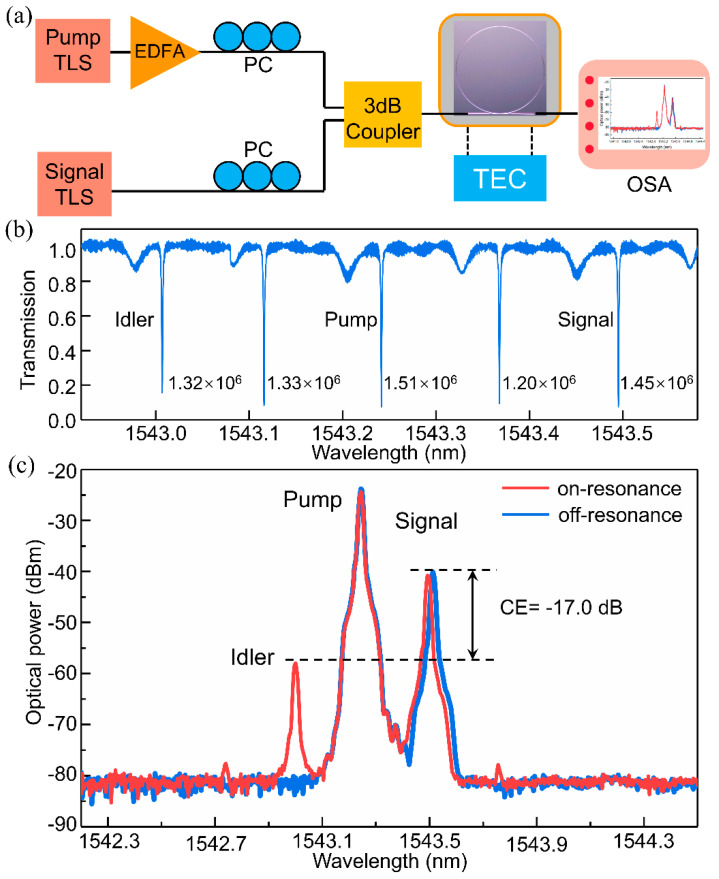
(**a**) The FWM measurement setup. (**b**) Under low input power, the transmission spectrum and Qs around the pump and signal wavelengths. (**c**) Measured highest conversion efficiency of the degenerate FWM.

**Figure 6 micromachines-13-01155-f006:**
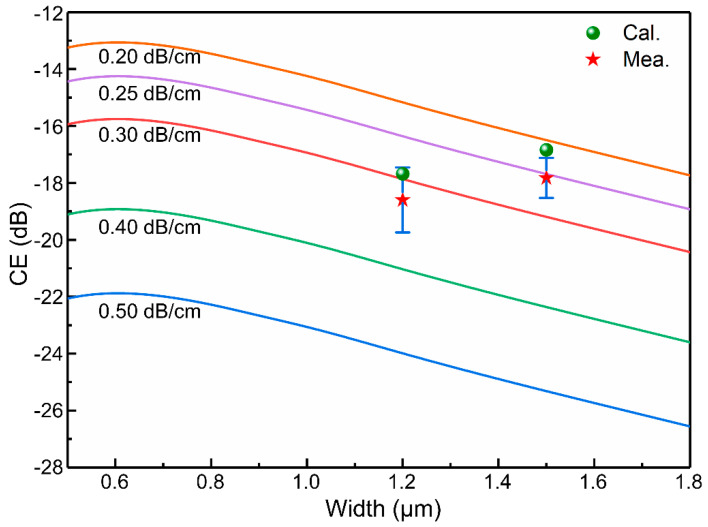
The solid lines represent the CE as a function of rib waveguides with different widths and different linear attenuation coefficients. Multimode rib MRRs can achieve greater CE owing to decreased propagation loss, as shown by the calculation results, which are depicted as dots. The stars represent the measured FWM CE of fabricated MRRs at different resonance peaks.

**Table 1 micromachines-13-01155-t001:** Comparison of degenerate FWM results for various types of SOI microrings and material platforms.

Device	Waveguide	Q-Factor	Pump Power (dBm)	CE (dB)	Reference
AlGaAs racetrack microring	Single mode	7.5 × 10^3^	13.80	−43	[38]
doped SiO_2_ add-drop microring	Single mode	1.2 × 10^6^	9.4	−26	[39]
amorphous SiC all-pass microring	Multimode	7 × 10^4^	11.76	−21	[40]
SOI all-pass microring	Single mode	9.15 × 10^3^	12	−29.4	[41]
SOI all-pass microring	Single mode	1.9 × 10^4^	10.5	−25.4	[42]
SOI cascaded microring	Multimode	3.0 × 10^4^	3	−22	[43]
SOI racetrack add-drop	Multimode	1.1 × 10^6^	10	−15.5	[44]
SOI all-pass microring	Multimode	1.51 × 10^6^	6.50	−17.0	This work

## Data Availability

Data underlying the results presented in this paper are not publicly available at this time but may be obtained from the authors upon reasonable request.

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
