# Peer review of "Demonstration of Ultra-High-Q Silicon Microring Resonators for Nonlinear Integrated Photonics"

_micromachines, 2022, doi:10.3390/mi13071155_

Round 1

Reviewer 1 Report

This manuscript demonstrates an ultra-high Q silicon ring resonator with high nonlinear conversion efficiency. The fabrication process is optimized to provide a low-loss waveguide. I would recommend accepting the paper with major revisions. Some necessary changes are outlined below.

1. Could the author add more reviews and comparisons on other techniques to achieve ultra-low loss waveguide? What is the advantage of current technology as compared to others?

2. A small suggestion regarding Eq.2. Could the author consider using numbers 0 and 1 for the parameters related to TE0 and TE1?

3. For Figure 2, could the author confirm that b&c and d&e are taken under the same condition? The noise and roughness in b and d seem related to the scan time/exposure time or poor focus of the SEM. This noise makes it hard to tell the difference between sidewall roughness. Also, could the author take the SEM with the wafer tilted to show the sidewall clearly? It is hard to get any solid conclusion with these SEMs.  

4. In line 134, the author used some experimental parameters to calculate the simulation result. The simulation perfectly matches the experiment. Could the author provide these values and how these values are calculated from the experimental result? If it is simply curve fitting, then there is no point discussing the perfect match.

5. In line 138, why is it worth noticing if the author does not use this method? I would recommend deleting it or moving it to the discussion and discussing more on this. In what case can it enable efficient conversion, and is it useful in the current experiment? Why is it not used in the current experiment?

6. In line 152, could the author provide how the loss is calculated?

7. In lines 156-158, the conclusion about the RP and oxidation smoothing does not relate to this paragraph. This paragraph discusses the waveguide width. The Q increase at a single point is also not convincing enough to prove that the oxidation smoothing case is better.  

8. In line 176, what does the “extinction ratio” refer to?

9. How does laser heating affect the high-power experiment?

10. In figure 6, since the width and loss are related, could the author also provide a figure to explain the loss-width relationship?

11. A careful check is required to clear grammar mistakes. 

Reviewer 2 Report

This paper fabricates an ultra-high Q-Factor microring resonator on Silicon using a rib waveguide. The fabrication process includes two innovations of Photoresist Reflow and Oxidation smoothing that help in substantially improving the resulting root-mean-square roughness of the fabricated structures leading to Improved Q-Factor of the microring and improved conversion efficiency for four-wave mixing.

But I have a few suggestions for the improvement of the manuscript:

1) The figure 1b needs to be redrawn with improved image quality as well as in such a way that the reader can clearly understand which dimension is for which part of the structure

2) The figure 1a needs to be redrawn with improved image quality and care must be taken that the words written are readable. For instance, t0, t1, t2, etc. are not properly readable. 

3) The image quality of all of the figures needs to be seriously improved except for the SEM images.

4) The authors mention oxidation smoothing starting from line 105. This is a general comment on oxidation smoothing, please also mention how was the oxidation smoothing performed for this study.

5) Language needs to be seriously improved in the passage between lines 145 and 158. The explanation given here is so haphazardly written that it is difficult to understand and follow. For instance, at line 149 it is said that "the average Qs of MRRs with 1.2 um and 1.5 um-wide waveguide are 1.17x10^6 and 1.2x10^6 respectively" then at line 153 it says "MRR with 1.5 um-wide waveguide, Qs larger than 1.3x10^6". This is one among many other examples in this part of the paper. 

6) Please clearly mention whether the results shown in Figure 6 are analytically calculated or through simulations?

7) Please re-write the passage from line 217 till 221 more clearly because at present the explanation for Figure 6 given here is difficult to comprehend.  

8) There are some typing mistakes as well, please correct them, for instance, "methon" on the line 203, "expolre" on the line number 234.

Reviewer 3 Report

The Authors report a microring resonator based on a multimode rib waveguide. The manuscript is well written and it deserves the publication after addressing the following contents:

1. The term Ultra-high Q factor is misleading. Several devices have been reported in literature with higher Q-factor (see, e.g. (2014). Integrated waveguide coupled Si 3 N 4 resonators in the ultrahigh-Q regime. Optica1(3), 153-157,  (2016, July). Rigorous model for the design of ultra-high Q-factor resonant cavities. In 2016 18th International Conference on Transparent Optical Networks (ICTON) (pp. 1-4). IEEE).

2. A discussion on materials to achieve high Q-factor should be reported. Losses of the order of dB/m have been reported in literature to achieve high Q factor (>10 ^6).

3. For the target application, the requirements should be reported.

Round 2

Reviewer 1 Report

The author addressed all my comments. I would like to recommend the publication of this paper.